# Automatic feature selection for performing Unit 2 of vault in wheel gymnastics

**Eiji Kitajima**[1]*, **Takashi Sato**[2], **Koji Kurata**[3], **Ryota Miyata**[3]*

**1** Graduate School of Engineering and Science, University of the Ryukyus, Nakagami, Okinawa, Japan,
**2** Department of Media Information Engineering, National Institute of Technology, Okinawa College, Nago, Okinawa, Japan, **3** Faculty of Engineering, University of the Ryukyus, Nakagami, Okinawa, Japan

* k218672@eve.u-ryukyu.ac.jp (EK); miyata26@tec.u-ryukyu.ac.jp (RM)

**Data Availability Statement:** All data are available at the following link: https://www.kaggle.com/datasets/aicoacheiji/wheel-gymnastics-vault-performance-dataset.

**Funding:** The authors received no specific funding for this work.

## Abstract

We propose a framework to analyze the relationship between the movement features of a wheel gymnast around the mounting phase of Unit 2 of the vault event and execution (E-score) deductions from a machine-learning perspective. We first developed an automation system from a video of a wheel gymnast performing a tuck-front somersault to extract the four frames highlighting its Unit 2 performance of the vault event, such as take-off, pike-mount, the starting point of time on the wheel, and final position before the thrust. We implemented this automation using *recurrent all-pairs field transforms (RAFT)* and *XMem*, i.e., deep network architectures respectively for optical flow estimation and video object segmentation. We then used a markerless pose-estimation system called *OpenPose* to acquire the coordinates of the gymnast's body joints, such as shoulders, hips, and knees then calculate the joint angles at the extracted video frames. Finally, we constructed a regression model to estimate the E-score deductions during Unit 2 on the basis of the joint angles using an ensemble learning algorithm called *Random Forests*, with which we could automatically select a small number of features with the nonzero values of feature importances. By applying our framework of markerless motion analysis to videos of male wheel gymnasts performing the vault, we achieved precise estimation of the E-score deductions during Unit 2 with a determination coefficient of 0.79. We found the two movement features of particular importance for them to avoid significant deductions: time on the wheel and angles of knees at the pike-mount position. The selected features well reflected the maturity of the gymnast's skills related to the motions of riding the wheel, easily noticeable to the judges, and their branching conditions were almost consistent with the general vault regulations.

## Introduction

Wheel gymnastics, or Rhönrad, is an acrobatic sports discipline that originated in Germany [1]. The gymnasts do exercises with a *wheel*, which consists of two rings connected by six bars. The judges award each gymnast D- and E-scores for technical difficulty and execution of the routine, respectively. Points are earned per skill to the D-score but deducted for any mistake from the perfect E-score of 6.0. The gymnast with the highest combined D- and E-score wins in the event competition.

**Competing interests:** No: The authors have declared that no competing interests exist.

Wheel gymnastics includes four competitive events: straight-line, spiral, vault, and cyr [2]. We focused on the vault, in which a gymnast attracts audience's attention with the dynamic performance of jumping onto the rolling wheel then jumping acrobatically from the top. Fig 1 shows an example of a *tuck-front somersault* addressed in our study. The somersault is an exercise in which a gymnast rotates forward with its legs bent after a strong jump from the final position before the thrust on the wheel. The general vault regulations [3] divide a vault performance from start to end into four units:

- Unit 1: Setting the wheel in motion, run-up (frames 1 to 6 in Fig 1)

- Unit 2: Take-off, mounting phase (frames 7 to 12)

- Unit 3: Thrust with hands or feet from the wheel, flight phase (frames 13 to 15)

- Unit 4: Landing (frames 16 to 18)

The judges start to evaluate a vault performance from Unit 2. In the mounting phase of Unit 2, it is difficult for a beginner to support its weight with its hands alone on the wheel, which is rolling forward. Hence, the judges tend to pay more attention to the skills of wheel gymnasts in Unit 2, which significantly affect the E-score. In other words, the key to victory in the vault of wheel gymnastics is not only succeeding in high D-score performance but

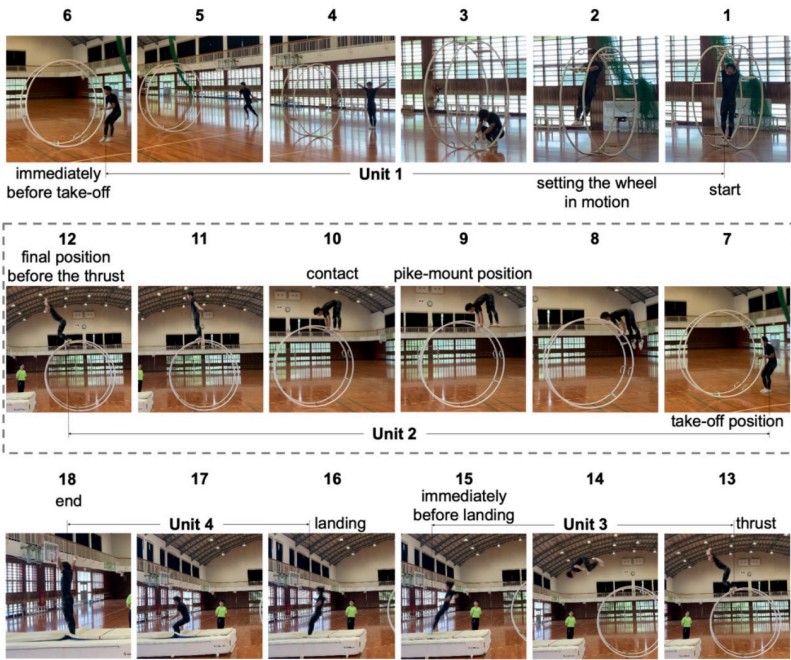

**Fig 1. Example of *tuck-front somersault*.** Before the vault event, a gymnast sets the wheel in motion to reach the landing mat with a maximum of two and a half rotations. After raising its hand to signal the start of its performance, he uses its weight to roll the wheel then runs up to it (Unit 1: frames 1 to 6). The video frames inside the dotted rectangle correspond to Unit 2, the subject of this study. After conducting the take-off from both feet and holding the rims of the wheel, the gymnast transitions to a pike-mount position (frames 7 to 9). The judges count from the gymnast's sole in contact with the wheel to the final position before the thrust as *time on the wheel* (frames 10 to 12). After a jump from the wheel (i.e., thrust), the gymnast rotates forward with its legs bent in the flight phase (Unit 3: frames 13 to 15). After landing on the mat in a stable position on both feet, he raises its arms to indicate that the vault performance has been completed (Unit 4: frames 16 to 18). The video of its vault performance can be viewed from the following link [4].

minimizing the E-score deductions during Unit 2, which includes many unique movements with a wheel not found in artistic gymnastics.

There are difficulties with visual scoring of Unit 2 performance. While most wheel gymnasts quickly complete the sequence of motions involved in a few seconds, judges must check the nearly eighty [3] review items according to the general vault regulations. Some might look into using sensors or other devices fixed to gymnasts or their gears to monitor their detailed movements. However, this methodology is unsuitable for sports-data collection because it often leads to physical and mental restrictions on the gymnasts' performances. In some sports, training at an elite level may involve using dedicated motion-capture systems with multiple calibrated cameras and markers placed on the player. However, such marker-based systems are impractical in lower-end training sessions and during actual competitions [5]. Therefore, the scorings depend heavily on the judges' experiences and impressions, which influence what motions they regard as targets for the deductions. The judges never give any breakdown of the scoring in the competitions.

Previous studies on wheel gymnastics have reported identifying areas of the body where gymnasts are prone to injury or suffer from overuse syndrome [6, 7] and assessing correlations between the body fat percentage, competitive results, and motivation [8, 9]. Certain studies have used a sonification system to improve the performance of a gymnast (e.g., [10]). However, to our knowledge, no studies have quantitatively addressed the execution of Unit 2 performances in the vault event, and few studies qualitatively analyzed the take-off motion from the viewpoint of biomechanics [11, 12]. Thus, the relation between the overall movements of Unit 2 by a wheel gymnast and E-score deductions is still unclear.

To solve the problems mentioned above, we developed a framework to analyze the relationship between the movement features of a wheel gymnast during Unit 2 of the vault event and the E-score deductions from a machine-learning perspective, as shown in Fig 2. Using deep-

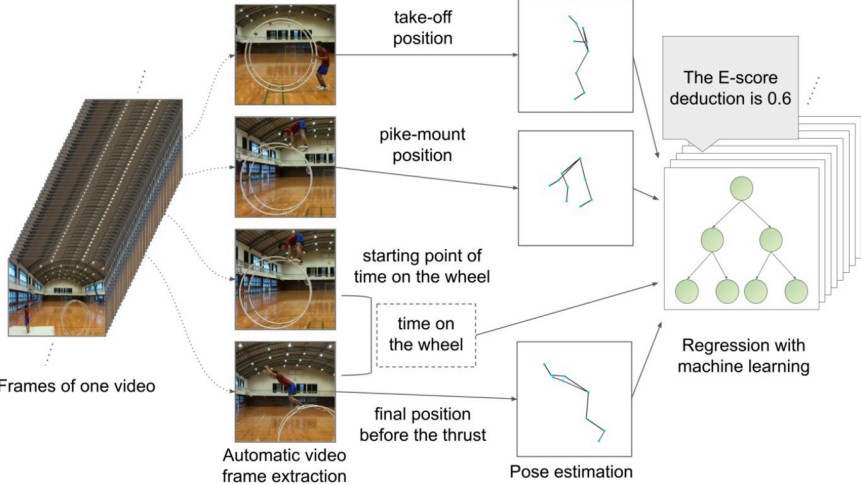

**Fig 2. Proposed framework for markerless motion analyses of Unit 2 in Category B vault performance.** First, from a performance video, we extract the four frames required for scoring the execution using deep learning-based models; RAFT [13] and XMem [14]. Second, from the extracted video frames, we obtain the joint coordinates of the on-screen gymnast using a pose-estimation model (BODY_25) of OpenPose [15] then calculate the joint angles in each frame. Finally, on the basis of the joint angles, we estimate the E-score deductions during Unit 2 using Random Forests [16], which enables us to automatize the feature selection by reference to the nonzero values of feature importances. More details of these three processes are in the fourth to sixth subsections of Materials and Methods (see also Figs 4–6). This figure was created using ML Visuals [17].

learning algorithms for optical flow estimation and video object segmentation, we first implemented automatic extraction of the video frames required for scoring the execution of Unit 2 from a video of the wheel gymnast performing a tuck-front somersault. We then adopted a markerless motion-capture technique called pose estimation to obtain the coordinates of its body joints in each extracted frame. We also constructed a tree-based ensemble regression model to estimate the E-score deductions during Unit 2 from the motion features on the basis of the obtained coordinates of the gymnast's joints. As well as high estimation accuracy, the model realizes automatic feature selection for avoiding the significant deductions by reference to the nonzero values of feature importances.

Our regression model enabled data-driven personal training for a wheel gymnast and their coaches, informing them of its unconscious bad habits during Unit 2 by submitting concrete numerical evidence.

## Materials and methods

Fig 2 shows the outline of our research. First, we recorded 108 videos of wheel gymnasts performing a tuck-front somersault then obtained the E-scores that two judges independently gave for each vault performance of Unit 2. Second, we applied the architecture recurrent all-pairs field transforms (RAFT) [13] and current state-of-the-art architecture with unified feature memory stores (XMem) [14] to the videos and automatically extracted the four frames involving Unit 2, i.e., take-off, pike-mount, the starting point of time on the wheel, and final position before the thrust. Third, applying OpenPose [15] to the extracted video frames, we estimated the coordinates of the gymnast's body joints and calculated its joint angles in each frame. Finally, using Random Forests [16], we constructed a regression model that estimated the E-score deductions in Unit 2 on the basis of the joint angles at the three poses and selected features associated with the significant deductions. The Random Forests automatically select a small number of motion features with nonzero feature importance (FI) values related to the E-score deductions.

### Movements to be analyzed: Unit 2 of Category B vault

The following are the reasons we chose the tuck-front somersault for motion analysis, as illustrated in Fig 1. This vault performance contains many fundamental movements in the vault event of wheel gymnastics, and certain intercollegiate championships adopt it as a compulsory exercise [18].

According to the general vault regulations [3], there are four different vault categories:

- Category A: Vaults performed in a tuck or straddle position from arm support on the wheel

- Category B: Vaults performed from a forward standing position on the wheel

- Category C: Vaults performed from a reverse standing position on the wheel

- Category D: Vaults performed using an overswing technique

A Category-A vault can be performed even by beginners whose muscles have not yet developed enough to support their weight with arm strength alone. Categories-C and -D vaults, however, require gymnasts to practice more advanced skills than executing Category B, such as crossing their legs to change their position from front-facing to rearward or swinging their legs back and up during the mounting phase.

Since the tuck-front somersault belongs to Category B, we focused on the gymnast's movement from the frontal ride on the wheel to just before the jump-out, as shown with the dotted line in Fig 1.

## Participants

There were four male participants in our study; two wheel gymnasts (gymnast 1 and gymnast 2) performed the tuck-front somersaults, and two judges independently gave the E-score deductions for each performance in Unit 2. The wheel gymnasts have similar heights and build, and they also have a similar level of technical proficiency in performing tuck-front somersaults in the vault. The individuals in this manuscript have given written informed consent (as outlined in PLOS consent form) to publish these case details. The Ethics Committee of the University of the Ryukyus approved this study (approval number 36 and 58).

## Data collection

We recorded 108 videos of performing tuck-front somersaults by the wheel gymnasts. These videos were the sources of inputs to the regression model for estimating the E-score deductions in Unit 2 described as follows. For gymnast 1, we recorded an average of 13 vault performances per day over eight days using two cameras (GoPro HERO7 Black [19], 60FPS), which we placed orthogonally to and 3.1 m from the left sideline of the run-up area. The first camera served to capture the take-off and pike-mount positions (Fig 3(a) and 3(b), respectively), while the second did so for the whole movements from the gymnast's feet landing on the wheel after the pike-mount position to the final position before the thrust (Fig 3(c), i.e., to the end of Unit 2). We set the second camera 1.1 m higher than the first to ensure that the hands raised by the gymnast standing on the wheel to take the final position before the thrust does not protrude from the video frames. The same recording conditions were used for gymnast 2 with an average of 3 recordings over three days. In this study, we used OneFormer [20] to mask the entire video except for the gymnasts in order to maintain anonymity when a non-gymnast person appears in the videos of camera 2. The same goes for camera 1 with the XMem-estimated masks. These process was conducted before the pose estimation. All data are available at the following link [21]. In analogy with judges in an official competition, we watched the vault performances from only one side of the wheel gymnast. Since Category B vaults do not require

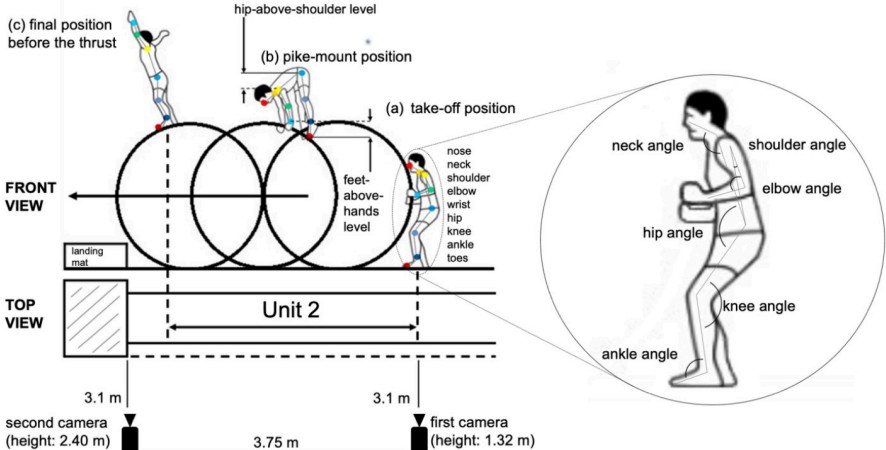

**Fig 3. Schematic representation of recording conditions for Unit 2 movements in vault performance with orthographic view.** We use OpenPose to obtain the two-dimensional (2D) coordinates of the nine joints on the left-side body of a wheel gymnast in each video frame. We then calculate the six joint angles of the gymnast on the basis of these coordinates and use them for inputs of the regression model for estimating the E-score deduction during Unit 2. Reprinted from [22] under a CC BY license, with permission from Sorao Kosaka, original copyright 2020.

any twisting movement in the mounting phase and there is almost no difference between left and right movements, we focused on the left-side of the gymnast's body.

Subsequently, we asked two volunteer judges with experience in judging national competitions to independently score each Unit 2 performance by the gymnast after they had watched the 108 video recordings of its tuck-front somersaults. Following the general vault regulations [3], they respectively made E-score deductions between 0.0 to 6.0 for any aesthetic or technical error during Unit 2. We used each arithmetic mean of the deductions produced by the two judges as a target of the regression model.

## Automatic extraction of video frames related to Unit 2 performance

To extract the specific frames necessary to evaluate the execution of Unit 2 from a video of a wheel gymnast performing a tuck-front somersault, we used two deep-learning-based image-recognition techniques: optical flow estimation and video object segmentation. The optical flow tells us which direction and how far objects in the video have moved per pixel, providing a vector field between two consecutive frames. For fast and precise optical flow estimation, we used the official implementation [23] of RAFT. By tracking changes in the optical flow estimated with RAFT, we detected two types of jump motions in the vault of wheel gymnastics: take-off and final position before the thrust. Video object segmentation aims to separate object regions of interest (ROIs) from the background in a given video. For high-quality object segmentation on videos, we used Colab Jupyter Notebook environment [24] provided by XMem. By calculating the barycenters and boundaries of ROIs, we detected the unique phases of Unit 2: pike-mount position and starting point of time on the wheel. The following details the frame-extraction methodologies.

- Take-off position
  When the median direction of the optical flow estimated with RAFT kept turning upward (0 [°] $< \theta <$ 180 [°]) for thirty consecutive frames (0.5 s), we extracted the preceding frame, the last one that had the downward flow direction (180 [°] $< \theta <$ 360 [°]), as that of take-off, as shown in Fig 4(A). We traced back the frame from that of take-off until the center_of_mass function [25] of SciPy library [26] separately detected two masks, one for the wheel and the other for the gymnast. By providing these masks as the first-frame annotations (i.e., target data) to XMem, we achieved full automation of semi-supervised object segmentation. XMem enabled us to focus only on the positional relation between the wheel and gymnast, removing the background image at each frame.

- Pike-mount position to the starting point of time on the wheel
  For detecting the pike-mount position, we first needed to determine the starting point of time on the wheel, i.e., the first frame of contact with the wheel by the gymnast's soles. To extract this video frame, we needed to avoid false detection of gripping the wheel with its hands in the take-off or by touching it with its shoulders/arms for a moment around the pike-mount position. We thus calculated the barycenter of the human region segmented with XMem using the center_of_mass function of the SciPy library and defined the lower body as a part to the right of the x-coordinate (i.e., blue area), as shown in Fig 4(D). We then clipped the first of the frames with more than 0.5 seconds of contact between XMem-segmented regions of the gymnast's lower body and the wheel. From the frames before this, we chose the one with the highest barycenter of the human region segmented with XMem as the pike-mount position, as shown in Fig 4(C).

- Final position before the thrust
  When confirming a change in the optical flow direction, i.e., the median at the current frame

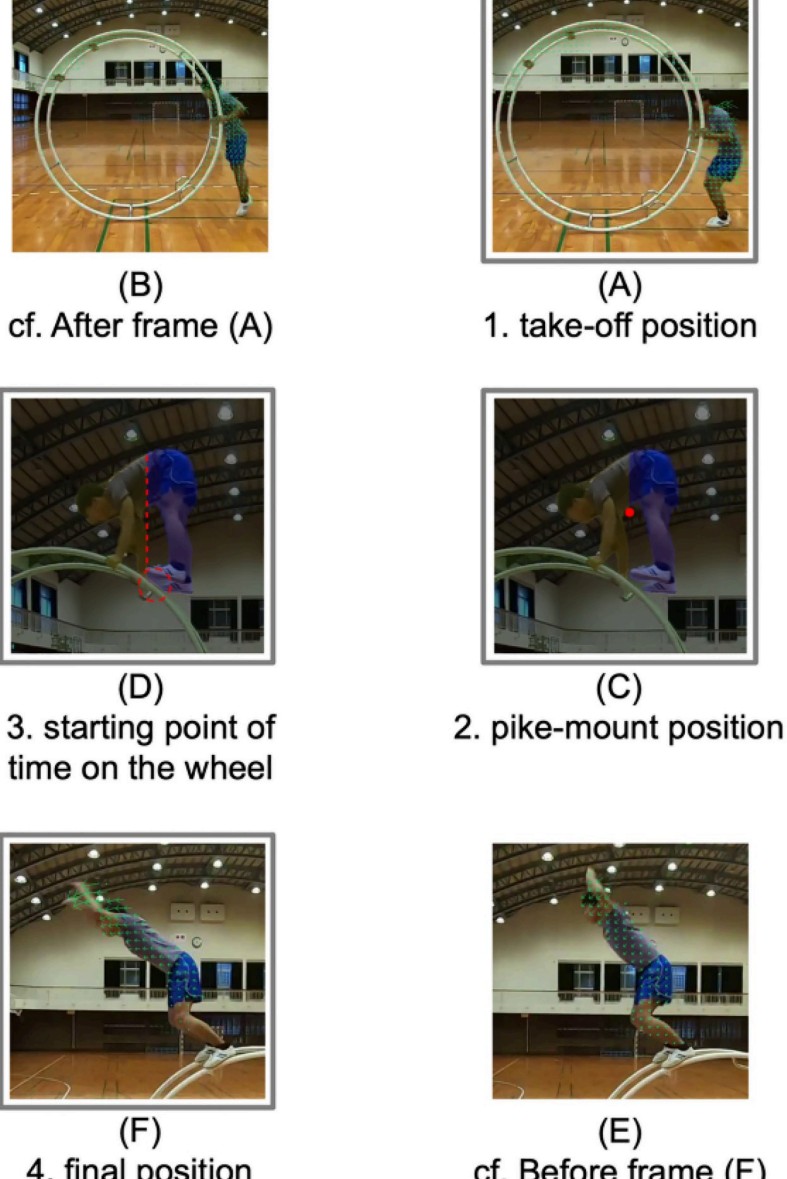

**Fig 4. Automatic extraction of four frames related to Unit 2 with RAFT and XMem from video.** In the upper and lower rows, the direction and length of each arrow represent the RAFT-estimated direction and magnitude of optical flow at each pixel of the frame. Extracting the last frame preceding the median direction of the optical flows pointing continuously upward due to (B) the jump, we could capture (A) take-off position. In the same way, extracting the first frame in which the median direction of the optical flows changed from continuously downward to upward after (E) the squat position, we could capture (F) the final position before the thrust (i.e., the endpoint of time on the wheel). In the middle row, the yellow and blue regions respectively represent the XMem-estimated masks of the wheel and the gymnast's lower body, which we calculated on the basis of the center of its segment mass displayed with the red dot. We regarded the first frame where these two masks coming into contact and displayed with the red line as (D) the starting point of time on the wheel. We set the frame with the maximum y-coordinate of the center of its mass preceding the starting point of time on the wheel as (C) the pike-mount position.

was upward and the ten frames (approximately 0.2 s) previously had been downward, we calculated the maximum median of flow magnitude during the ten frames. If the magnitude was the threshold of between 5.5 and 15, which we set to distinguish from the landing scene in Unit 3, we then extracted the corresponding frame as that of the final position before the thrust, as shown in Fig 4(F).

## Pose estimation

There are two well-accepted solutions for pose estimation: BlazePose [27] and OpenPose [15]. We adopted OpenPose to acquire the coordinates of the gymnast's body joints at the extracted video frames because preliminary experiments also confirmed that BlazePose had more instances where keypoints deviated from anatomical joint centers, as reported in the previous study [28].

We obtained the nine keypoints in the BODY_25 model of OpenPose: the nose, neck, shoulder, elbow, wrist, hip, knee, ankle, and toes, as shown in Fig 3(a). We use the following implementation [29]. Although OpenPose can detect both left and right-side joints of the human body, we adopted only the left-side ones as explanatory variables in regression for estimating the E-score deduction to reduce the number of features.

Fig 5 shows examples of pose estimation of a gymnast at each frame using OpenPose. As shown in the figure, we confirmed OpenPose successfully put the keypoints almost exactly on each body joint of the gymnast, and there were no missing them in our total 432 extracted video frames. If any keypoint were incidentally missing in the new data, we would discard them.

As shown in Fig 3, we calculated the body-joint angles using three body-joint coordinates. For instance, the coordinates of "neck," "hip," and "knee" yielded the "hip" angle. The details are illustrated in Fig 5.

## Random Forests

We used Random Forests [16] to estimate the E-score deductions for the gymnast's Unit 2 performance on the basis of its joint coordinates estimated using OpenPose in each of the three frames extracted with RAFT and XMem: take-off, pike-mount, and final position before the thrust.

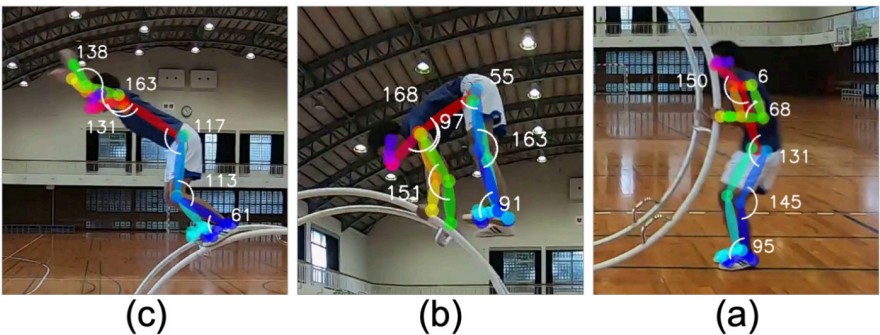

**Fig 5. Pose estimation with OpenPose on RAFT/XMem-extracted frames.** (a) take-off, (b) pike-mount, and (c) final position before thrust. Using OpenPose, we estimated the x- and y-coordinates of the wheel gymnast's nose, neck, shoulders, elbows, wrists, hips, knees, ankles, and toes, displayed in the colored points from top to bottom. By computing the arc cosine of two vectors given the coordinates of three adjacent joints, we obtained the angles of its neck, shoulder, elbow, hip, knee, and ankle, displayed with degrees. For example, to calculate the shoulder angle, we used the coordinates of its left elbow, shoulder, and hip.

**Table 1. Explanatory variables of Random Forests for estimating E-score deductions of Unit 2.**

| Frame | Take-off | Pike-mount | Before thrust |
|---|---|---|---|
| Variable Name [Unit Symbol] | Neck [°] | Neck [°] | Neck [°] |
| | Shoulder [°] | Shoulder [°] | Shoulder [°] |
| | Elbow [°] | Elbow [°] | Elbow [°] |
| | Hip [°] | Hip [°] | Hip [°] |
| | Knee [°] | Knee [°] | Knee [°] |
| | Ankle [°] | Ankle [°] | Ankle [°] |
| | | Hips-above-shoulder level [px] | |
| | | Feet-above-hands level [px] | |
| | | | Time on wheel [s] |

As shown in Table 1, we used 21 explanatory variables on the basis of the estimated joint coordinates. The first six rows of Table 1 represent the joint angles calculated from the joint coordinates in each frame. For example, we obtained the gymnast's shoulder angle in the take-off position by using the coordinates of its elbow, shoulder, and hip at the corresponding frame (see Fig 3 for details). *Hips-above-shoulders level* stands for the shoulders-to-hips height in the pike-mount position, and we calculated this by subtracting the y-coordinate of the shoulder from that of the hip at that frame. The same goes for *feet-above-hands level. Time on the wheel* refers to the duration of the gymnast's shoe soles in contact with the wheel before the thrust and is an indicator of whether the gymnast could execute its performance from riding the wheel to jumping it out without delay [3].

To compute the time on the wheel, we needed to find the starting and end points in advance, i.e., the sequence numbers of the first XMem-extracted frame at which the gymnast's soles are in contact with the wheel and those of the RAFT-extracted one corresponding to the final position before the thrust, as shown in Fig 4(D) and 4(F), respectively. We then obtained time on the wheel by dividing the difference between the sequence numbers of these two frames by the FPS of the recording camera.

The target (or objective) variable is the average of E-score deductions given independently by the two judges for the Unit 2 performance.

We suspected that since the condition of every athlete depends on the day, naively random split cross-validation might implicitly teach it to the model. To prevent such data leakage [30], we used sequential-split validation dividing the entire data set into the last two days' recordings (26 performances) for testing and the others (74 performances) for training. We used the root mean squared error (RMSE) and coefficient of determination $R^2$ between the model estimates and target values as the evaluation indices. We trained the Random Forests using 74 performances by gymnast 1, then validated them using his 26 latest ones and those 8 of gymnast 2.

We implemented the above tree-based ensemble regression model using RandomForestRegressor [31] of Scikit-learn [32]. For automatic feature selection to avoid significant E-score deductions, we measured each feature importance with the threshold of 0.1 of the trained model, i.e., the mean and standard deviation of accumulation of the impurity decrease within each tree. We then visualized the decision tree generated from the training and statistically derived rules on what the gymnast should pay special attention to in Unit 2 performance.

## Results

First, we implemented a Random-Forests regression model to estimate the E-score deductions for the wheel gymnast performing Unit 2 of Category-B vault on the basis of the features of its

movements extracted using OpenPose and evaluated the test accuracy of the regression model. Fig 6(A) is the scatter plot of the actual values versus the estimated ones obtained through sequential-split validation. Our estimations based on the body-movement features strongly correlated with the actual deductions by the judges ($R^2 = 0.79$) and had a relatively small margin RMSE = 0.14. This suggests that the model's predictions are applicable for a gymnast with a similar level of technical proficiency in tuck-front somersaults.

For comparison, we also estimated the E-score deductions using Long short-term memory (LSTM) networks for time-series data [33], feeding them the same sequential inputs as the Random Forests. As shown in S1 Fig, the LSTM only achieved lower accuracies than the Random Forests regarding either $R^2$ or RMSE.

Second, we investigated the feature importance (FI) for estimating the E-score deductions: the top 7 are listed in Fig 6(B). The first was time on the wheel (FI = 0.798) and the second was the knee angle in the pike-mount position (FI = 0.161). For the third and subsequent features, all values were nearly zero. We found that the top two features with the highest values of FI contributed significantly to predict the E-score deductions while the third and subsequent ones were close to zero.

Finally, we demonstrated the decision-tree rules produced by training the Random-Forests regression model and illustrate two examples in Fig 6(C) and 6(D). The split is based on the criterion of the mean squared error. When looking at the path with the fewest E-score deductions of 0.36 estimated from the decision tree in Fig 6(C), we see that the time on the wheel did not exceed 4.935 s, and the left knee angle in the pike-mount position was more than 162.5˚. Fig 6(C) also shows that the decision tree recognized the case when the time on the wheel was longer than 6.07 s as the target for the most significant deductions. Shortening the time on the

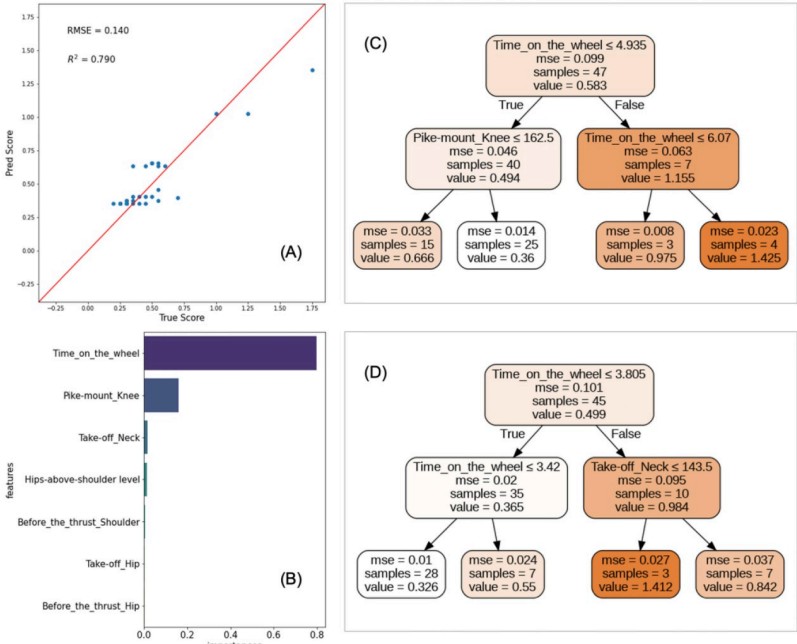

**Fig 6. Regression of E-score deductions during Unit 2 of Category B vault using Random Forests.** (A) Scatter plot of the estimated deductions and true ones. (B) Top 7 feature importances in Random Forests. (C), (D) Examples of the decision trees composing Random Forests. Each value of the nodes in Fig 6(C) and 6(D) represents the E-score deductions predicted by the regression model. The explanatory variables of regression were shown in Table 1. Note that the values of feature importance for the 8th and subsequent features were all zero.

wheel was the key to minimizing the E-score deductions during Unit 2, and the same was true for Fig 6(D), which also required consideration of neck angle in the take-off position. The diagram indicates that the gymnast would obtain the deductions of 0.842 if the time on the wheel is not later than 3.805 s and if he could keep its neck angle at least 143.5˚ in the take-off position.

## Discussion

We proposed a framework to analyze the relationship between the movement features of a wheel gymnast during Unit 2 of the vault event and the E-score deductions from a machine-learning perspective. Using deep-learning algorithms for optical flow estimation and video object segmentation, we first implemented automatic extraction of the video frames required for scoring the execution of Unit 2 from a video of the wheel gymnast performing a tuck-front somersault. To the best of our knowledge, this is the first attempt to introduce computer vision technologies for quantitative evaluation of the vault performance of wheel gymnastics. Although we developed our framework to analyze the videos of a wheel gymnast moving from right to left, we can also apply it to those in the opposite direction because it captured the positional relations between the segmented objects and changes in the estimated flow directions regardless of which way to move.

We then introduced the markerless pose-estimation model to precisely capture the motions of the wheel gymnast in Unit 2. It enables the gymnast to perform the vaults in the same format as in an actual competition without the constraints of wearing any sensors or devices during the recording sessions. As far as we know, this study achieved the world's first digitalization of a wheel gymnast's vault performances, such as its knee extension at the moment of the mounting phase. We used a 2D pose-estimation model, which captured the horizontal and vertical motions of the gymnast, to minimize the computational costs of machine learning for simplicity. For future analyses that include body twisting movements such as Category C vault, it is necessary to introduce a 3D extended model that can also estimate the depth directions of human joint coordinates.

The proposed framework revealed the gymnast's motions which led to significant E-score deductions during Unit 2 by applying an ensemble-tree-based machine-learning algorithm to the body joint data obtained from the pose-estimation model. Following this framework, a wheel gymnast would find issues to address, i.e., the skills he could demonstrate only when in good shape. If a gymnast looks at the branching conditions of each decision tree shown in Fig 6 (e.g., a performance that he spent on the wheel for more than 6 s results in a severe deduction of 1.425), he can practice mitigating these issues.

We achieved a high-accuracy estimation of the total E-score deduction on the basis of body joint data with an RMSE = 0.14 and $R^2$ = 0.79. For reference, in the vault during artistic gymnastics competitions, it is believed that there is a mean deviation of 0.66 from the reference score when scoring the performance [34]. In our case, the RMSE between the Random Forests estimation and average of E-scores given by the experienced judges was smaller than that of the reference value. Therefore, the estimation accuracy of this model is considered reasonable.

Using hypothesis tests, we also verified whether the conditional branches appearing in each decision tree of the Random Forests make statistically significant differences in the E-score deductions. First, we looked at the conditional branches in Fig 6(C) that would minimize the deductions when both were satisfied: time on the wheel was within 4.935 s, and knee angle in the pike-mount position was over 162.5˚. Dividing the vault performances into four groups by the combination of the time $t$ and knee angle $\theta_{knee}$ conditions shown in Fig 7(A), we conducted a multiple-comparison test among them on the average E-score deductions produced

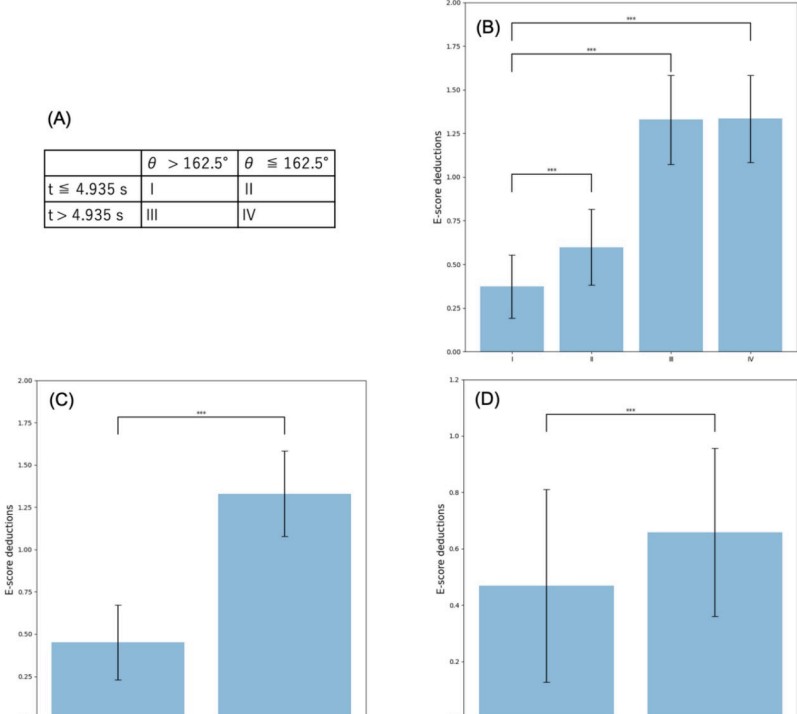

**Fig 7. Statistical hypothesis tests on conditional branches to minimize E-score deductions in Fig 6(C).** (A) Four groups of the vault performances divided by whether they met the two conditions derived by Random Forests: time on the wheel within 4.935 s, and knee angle in the pike-mount position over 162.5˚. (B) Multiple comparisons of the E-score deductions for the vault performances when (I) $t \leq 4.935$ s and $\theta_{knee} > 162.5$˚ with those of the other conditions (II)—(IV) (Dunnett's test [35, 36], $p < 0.001$). (C) Comparison of the E-score deductions between two groups divided only by whether $t \leq 4.935$ s or not (Mann–Whitney $U$ test [37, 38], $p < 0.001$). (D) Two-group comparison when $\theta_{knee} > 162.5$˚ and not (Mann-Whitney $U$ test, $p < 0.001$).

by the judges. As shown in Fig 7(B), employing Dunnett's test [35, 36] revealed the E-score deductions for a vault performance that met both conditions were statistically significantly smaller than the others ($p < 0.001$). We also compared the E-score deductions between two groups of the vault performances when $t \leq 4.935$ s and not regardless of $\theta_{knee}$. As can be seen in Fig 7(C), this time threshold statistically worked even alone for avoiding the significant deductions (Mann–Whitney $U$ test [37, 38], $p < 0.001$). Furthermore, as shown in Fig 7(D), the two-group comparison of the deductions divided only by whether the performance met $\theta_{knee} > 162.5$˚ yielded a significant statistical difference (Mann–Whitney $U$ test, $p < 0.001$), too. These statistical-analysis results support our claim from the proposed machine-learning methods because $t$ and $\theta_{knee}$ had the top-two feature-importance values shown in Fig 6(B).

Second, we looked at the conditional branches in Fig 6(D) that would maximize the E-score deductions when neither time on the wheel within 3.805 s nor greater than 143.5-degree neck angle in the take-off position was satisfied. S2(A)–S2(C) Fig display similar findings as in Fig 7: Dunnett's test revealed the deductions for a vault performance without the two conditions were statistically significantly larger than the others ($p < 0.001$, S2(A) and S2(B) Fig). Mann–Whitney $U$ test did time on the wheel statistically worked even alone for avoiding the significant deductions ($p < 0.001$, S2(C) Fig). On the other hand, no statistically significant differences were observed in S2(D) Fig comparing the deductions between the performances when the neck angle in the take-off position was less than 143.5˚ and not (Mann–Whitney $U$ test,

$p > 0.05$). As shown in Fig 6(C), compared to time on the wheel and knee angle in the pike-mount position, the feature-importance value of the neck angle in the take-off position was close to zero, which meant the node impurity hardly decreased by the feature of the neck angle alone. The neck-angle condition seemed to have an impact on the deductions in a complementary manner with time on the wheel because the Random Forests referred to it only when the performance could not satisfy $t \leq 3.805$ s in Fig 6(D). As demonstrated above, the results of statistical hypothesis tests supported those of the Random Forests in our proposed methods.

As shown in Fig 6(B) and 6(C), time on the wheel contributed most to estimating the total E-score deduction during Unit 2. Due to the characteristics of the apparatus, a wheel gymnast has to perform a vault on unstable footholds. Therefore, how trouble-free its performance is on the moving wheel will make a big impression on the judges. In other words, the judges tend to value the transition smoothness of the gymnast from getting on the wheel to the thrust. The general vault regulations [3] also state that the longer the time on the wheel is, the more points the gymnast will lose:

- 3–4 seconds: 0.1–0.2 deduction

- 5–6 seconds: 0.3–0.4 deduction

- > 6 s: 0.5 deduction

Looking at the branching conditions of the decision trees in Fig 6(C) and 6(D), their thresholds were very close to those of the general vault regulations, i.e., 4 and 3 s.

When we applied a single regression model with the time on the wheel as input and the total E-score deduction for the vault performance as output, the RMSE and $R^2$ scores were only 0.20 and 0.65. These results suggest there are other features of the gymnast movements that affect the vault performance besides the time on the wheel, supporting Fig 6(B)–6(D).

Let us explain how critical the knee opening is in the mounting phase, referring to the decision tree in Fig 6(C). Fig 8(A) and 8(B) show examples of the knee angles in the pike-mount

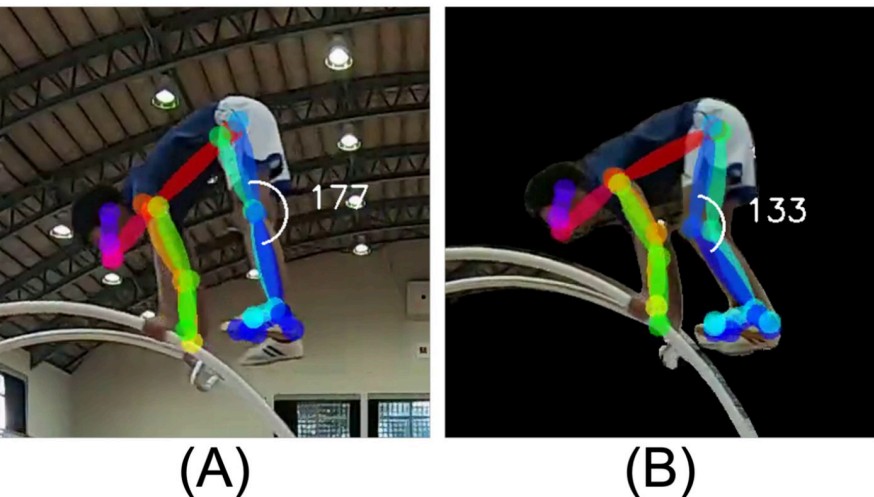

**Fig 8. Comparing (A) small and (B) large E-score deductions in pike-mount position.** In agreement with the decision-tree rule shown in Fig 6(C) and 6(D), the deductions varied depending on if the knee angle at this movement exceeded the threshold of 162.5˚. According to the general vault regulations [3], the hips and feet should be higher than the shoulders and hands level, respectively. The legs should also be straight. Straight legs are considered to have a greater impact on performance evaluation than the hips and feet levels. Therefore, the knee angle appeared as an important feature in Random Forests.

position when the total E-score deductions during Unit 2 were small and large, respectively. The former met the threshold of the left branch in Fig 6(C), i.e., knee angle of more than 162.5˚, while the latter did not. Comparing these two snapshots, one of the visible differences was whether the knees of the gymnast were straight or bent. The general vault regulations [3] define the ideal pike-mount position of the lower body as "closed hip angle with hips rising to above shoulder level, feet above the level of hands-on wheel, and legs straight throughout". Therefore, the bent knees could be a reason for significant E-score deduction since they stood out worse than the inadequate heights of hips and feet.

There were limitations to our study. We selected one gymnast for the model training with the ensemble-tree-based machine learning in a demonstration to reveal the challenges he was facing using our proposed framework. We thought statistical analysis of several gymnasts at once would average out the individual characteristics of motions, making it difficult to determine the issues each gymnast should address. As mentioned above, all the movement features that appeared as branching conditions of the decision trees in this analysis were easily noticeable to the judges and reasonable to interpret. However, we cannot say for sure that the results are true for other gymnasts as well. If other gymnasts incorporate the proposed framework into their practice and we aggregate the results of their motion analyses, it may be possible to identify the common issues that wheel gymnasts tend to encounter at each step of skill acquisition. Moreover, we employed Random Forests for estimating the E-score deductions from the motion features instead of LSTM networks developed for handling sequence data due to achieving high estimation accuracy and model interpretability with the small data collected under the circumstances in which both the numbers of performances done per day and subjects who could execute them were limited. Given thousands of video recordings enough to optimize the LSTM parameters, a sophisticated model may allow a more detailed examination of the sequential movements of riding the wheel, which will be our future work.

## Supporting information

**S1 Fig. Scatter plot of the E-score deductions estimated using LSTM network instead of Random Forests.** This model consisted of an input layer with 21 units, an LSTM layer with 64 internal ones, and a dense layer of 1 output with the identity function (for more details of the hyperparameters, see S1 Table). The inputs of training and test data were identical to those in the main body of our manuscript, with standardization.
(TIF)

**S2 Fig. Statistical hypothesis tests on conditional branches to maximize E-score deductions in Fig 6(D).** (A) Four groups of the vault performances divided by whether they met the two conditions derived by Random Forests: time on the wheel over 3.805 s, and neck angle in the take-off position within 143.5˚. (B) Multiple comparisons of the E-score deductions for the vault performances when (I) $t > 3.805$ s and $\theta_{neck} \leq 143.5$˚ with those of the other conditions (II)—(IV) (Dunnett's test, $p < 0.001$). (C) Comparison of the E-score deductions between two groups divided only by whether $t > 3.805$ s or not (Mann–Whitney $U$ test, $p < 0.001$). (D) Two-group comparison when $\theta_{neck} \leq 143.5$˚ and not (Mann-Whitney $U$ test, $p \geq 0.05$).
(TIF)

**S1 Table. Hyperparameters of LSTM network in S1 Fig.**
(TIF)

## Acknowledgments

We are grateful to Mr. Koji Hayashi and Mr. Yoji Uechi of the University of the Ryukyus gymnastics team for their help in assigning training labels for the vault. We would like to thank Sorao Kosaka for providing a clear diagram (Fig 3).

## Author Contributions

**Conceptualization:** Eiji Kitajima, Takashi Sato, Ryota Miyata.

**Data curation:** Eiji Kitajima.

**Formal analysis:** Eiji Kitajima.

**Investigation:** Eiji Kitajima.

**Methodology:** Eiji Kitajima, Ryota Miyata.

**Project administration:** Eiji Kitajima, Takashi Sato, Koji Kurata, Ryota Miyata.

**Resources:** Eiji Kitajima.

**Software:** Eiji Kitajima.

**Supervision:** Ryota Miyata.

**Validation:** Eiji Kitajima, Ryota Miyata.

**Visualization:** Eiji Kitajima.

**Writing – original draft:** Eiji Kitajima, Ryota Miyata.

**Writing – review & editing:** Eiji Kitajima, Takashi Sato, Koji Kurata, Ryota Miyata.

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
