## [Decision Letter · Decision Letter 0]

8 Feb 2023

PONE-D-22-33476Feature selection for performing Unit 2 of vault in wheel gymnasticsPLOS ONE

Dear Dr. KITAJIMA,

Thank you for submitting your manuscript to PLOS ONE. After careful consideration, we feel that it has merit but does not fully meet PLOS ONE’s publication criteria as it currently stands. Therefore, we invite you to submit a revised version of the manuscript that addresses the points raised during the review process.

We look forward to receiving your revised manuscript.

Kind regards,

Noman Naseer, PhD

Academic Editor

PLOS ONE

Journal Requirements:

"NO: The funders had no role in study design, data collection and analysis, decision to publish, or preparation of the manuscript."

4. We note that Figure 3 in your submission contain copyrighted images. All PLOS content is published under the Creative Commons Attribution License (CC BY 4.0), which means that the manuscript, images, and Supporting Information files will be freely available online, and any third party is permitted to access, download, copy, distribute, and use these materials in any way, even commercially, with proper attribution. For more information, see our copyright guidelines: http://journals.plos.org/plosone/s/licenses-and-copyright.

a. You may seek permission from the original copyright holder of Figure 3 to publish the content specifically under the CC BY 4.0 license. 

5. We note that Figures 1, 2 and 7 includes an image of a participant in the study. 

Additional Editor Comments:

Associate Editor: Reviewers have found potential in the study. However, there are several concerns that need to be addressed and I concur with the reviewers. I encourage the authors to revise upon reviewers comments and resubmit the manuscript.

Reviewers' comments:

Reviewer's Responses to Questions

**Comments to the Author**

1. Is the manuscript technically sound, and do the data support the conclusions?

Reviewer #1: Yes

Reviewer #2: Yes

2. Has the statistical analysis been performed appropriately and rigorously? 

Reviewer #1: Yes

Reviewer #2: Yes

3. Have the authors made all data underlying the findings in their manuscript fully available?

Reviewer #1: No

Reviewer #2: Yes

4. Is the manuscript presented in an intelligible fashion and written in standard English?

Reviewer #1: Yes

Reviewer #2: Yes

5. Review Comments to the Author

Reviewer #1: In this study, authors proposed a framework to analyze the relationship between the movement features of a wheel gymnast during Unit 2 of the vault event and the E-score deductions from ML.

o Authors mentioned deep network architectures in the abstract. Better to mention the name of the deep learning architecture.

o Authors used deeplabcut to acquire the coordinates of the gymnast's body joints. Nowadays openpose is quite popular for the extraction of body joints. Openpose also estimates based on optical flow.

Noori, Farzan Majeed, et al. "A robust human activity recognition approach using openpose, motion features, and deep recurrent neural network." Scandinavian conference on image analysis. Springer, Cham, 2019.

Cao, Z., et al. "OpenPose: Realtime Multi-Person 2D Pose Estimation Using Part Affinity Fields. In IEEE Transactions on Pattern Analysis and Machine Intelligence." (2019).

o If some joints of the frames were missing, how authors incorporated with such sort of issue, which is common in body joints detection through computer vison.

o It is time series data, why authors preferred random forests why not state-of-the LSTMs, in that case it would be end-to-end deep learning approach.

o Authors mentioned *we extract the four frames required for scoring the execution using deep learning-based computer-vision techniques* . Please specify the model/technique here or mention the section number about further details, it was a bit challenging to follow.

o Title of the manuscript shows authors mention selection of features. I am failing to understand that the selection was manual or automatic.

o Abstract line 5: it should be its not his. Please proofread the whole manuscript.

Reviewer #2: The authors of the manuscript present a framework for analyzing the relationship between the movement features of a wheel gymnast during the mounting phase of Unit 2 of the vault event and execution (E-score) deductions, using a machine learning approach. They utilized gymnastics rules and machine learning techniques to determine the E-score deductions. According to the authors, this is the first study of its kind to quantify the E-score deductions in wheel gymnastics using a computer vision approach.

Some areas for improvement in the manuscript include increasing the number of subjects, as the results of the proposed framework would be stronger if they were based on the analysis of multiple gymnasts, allowing for statistical analysis and the application of null hypothesis to assess the significance of the method. The quality of the figures, specifically Figure 6, could also be improved by enlarging the font size of the x- and y-axes for better readability. Additionally, the manuscript's sentence structure and use of academic-style English could be improved.

6. PLOS authors have the option to publish the peer review history of their article (what does this mean?). If published, this will include your full peer review and any attached files.

Reviewer #1: **Yes: **Farzan Majeed Noori

Reviewer #2: **Yes: **Hammad Nazeer

---

## [Author Response · Author response to Decision Letter 0]

30 Mar 2023

Academic Editor comments

Thank you for submitting your manuscript to PLOS ONE. After careful consideration, we feel that it has merit but does not fully meet PLOS ONE’s publication criteria as it currently stands. Therefore, we invite you to submit a revised version of the manuscript that addresses the points raised during the review process.

[Our response] Thank you for the positive comments about our manuscript. We will do our best to get it published.

Journal Requirements

[Our response] We have checked that our manuscript meets the PLOS ONE’s style requirements. We have also listed Supporting Information captions at the end of the manuscript in a section titled “Supporting information” (p. 13, line 484 – line 491).

"NO: The funders had no role in study design, data collection and analysis, decision to publish, or preparation of the manuscript."

[Our response] The authors received no specific funding for this work. We have included our amended statements within our revised cover letter as “The authors received no specific funding for this work.”

[Our response] We have included the Data Availability statement within our revised cover letter as “All data are available at the following link. 

https://drive.google.com/drive/folders/1eQpiWy6cjM07_QN6w5bXC8CLez02pzY1?usp=sharing”.

4. We note that Figure 3 in your submission contain copyrighted images. All PLOS content is published under the Creative Commons Attribution License (CC BY 4.0), which means that the manuscript, images, and Supporting Information files will be freely available online, and any third party is permitted to access, download, copy, distribute, and use these materials in any way, even commercially, with proper attribution. For more information, see our copyright guidelines: http://journals.plos.org/plosone/s/licenses-and-copyright.

a. You may seek permission from the original copyright holder of Figure 3 to publish the content specifically under the CC BY 4.0 license. 

[Our response] We have obtained permission from the original copyright holder of Figure 3 to release the content under a CC BY 4.0 license in accordance with the above requirement (1), a, and uploaded the completed Content Permission Form as an "Other" file. We have also included the following text in our revised manuscript: “Reprinted from [22] under a CC BY license, with permission from Sorao Kosaka, original copyright 2020” in the figure 3 caption (p. 5, after line 153).

5. We note that Figures 1, 2 and 7 includes an image of a participant in the study. 

[Our response] The specific permission to publish under the PLOS open-access (CC-BY) license has been obtained from the participants in this study. We have included the following text in the revised manuscript: The individuals in this manuscript have given written informed consent (as outlined in PLOS consent form) to publish these case details (p. 4, lines 128-130).

Additional Editor Comments:

Associate Editor: 

Reviewers have found potential in the study. However, there are several concerns that need to be addressed and I concur with the reviewers. I encourage the authors to revise upon reviewers comments and resubmit the manuscript.

[Our response] Thank you for your suggestion. We took the reviewers' comments seriously and incorporated them into the manuscript to the best of our ability.

5. Review Comments to the Author

Reviewer #1: In this study, authors proposed a framework to analyze the relationship between the movement features of a wheel gymnast during Unit 2 of the vault event and the E-score deductions from ML.

o Authors mentioned deep network architectures in the abstract. Better to mention the name of the deep learning architecture.

[Our response] Following your suggestion, we added RAFT, the abbreviated name of deep learning architecture, to line 9 of the abstract (p. 1). 

We also italicized the names of deep or machine learning algorithms used in this study to make them stand out.

o Authors used deeplabcut to acquire the coordinates of the gymnast's body joints. Nowadays openpose is quite popular for the extraction of body joints. Openpose also estimates based on optical flow.

Noori, Farzan Majeed, et al. "A robust human activity recognition approach using openpose, motion features, and deep recurrent neural network." Scandinavian conference on image analysis. Springer, Cham, 2019.

Cao, Z., et al. "OpenPose: Realtime Multi-Person 2D Pose Estimation Using Part Affinity Fields. In IEEE Transactions on Pattern Analysis and Machine Intelligence." (2019).

[Our response] To ensure the generalization of pose estimation we changed the pose estimation model from DeepLabCut to OpenPose. Previously, we avoided using OpenPose because we mistakenly believed that it could not be used in the sports field due to the following license. "The non-exclusive commercial license cannot be used in the field of Sports.", according to this page. However, we found the following text on the same page: "Just looking for an academic license?" which confirms that there is a license for academic use only. We have reflected this comment in the revised manuscript (p. 6, lines 209 - 218) and in the figure 5 caption (p. 7, after line 226).

o If some joints of the frames were missing, how authors incorporated with such sort of issue, which is common in body joints detection through computer vison.

[Our response] There was no missing detection of body joints in this study, and if there had been, we would have discarded the data. The possibility of missing detection of body joints is small because we carefully selected the recording conditions and placed the cameras at distances that would allow us to capture the gymnasts. We have reflected this comment in the revised manuscript (p. 6, line 220 and p.7, lines 221 - 223).

o It is time series data, why authors preferred random forests why not state-of-the LSTMs, in that case it would be end-to-end deep learning approach.

[Our response] Before the first submission of our manuscript, we had also considered an end-to-end deep-learning approach in which the inputs of the LSTM network to estimate the E-score deductions were the time-series data of each body joint coordinate obtained with a pose estimation model. This approach, however, would require at least thousands of performance videos since the time on the wheel could fluctuate at each performance, and it was impractical to acquire such an enormous dataset.

To select the significant features of motions related to the E-score deductions from a limited number of data, we first focused on the critical phases mentioned in the general vault regulations by implementing the automatic extraction of the video frames required for the scoring. 

We then narrowed down the candidates by reference to the value of feature importance output from the decision-tree-based ensemble model, which did not require thousands of samples or preprocessing like normalization and standardization that deep learning did.

For comparison, we included in the Supporting Information of our manuscript the results of estimating the E-score deductions using an LSTM network input with the same 21 features as those of the random forest and trained with the same training data.

As shown in S1 Figure, the LSTM only achieved lower accuracies than the Random Forests in terms of either R2 or RMSE.

We have reflected the above comments in the revised manuscript (p. 8, lines 276 - 278) and in the S1 Fig caption (p. 13, lines 486 - 490).

o Authors mentioned *we extract the four frames required for scoring the execution using deep learning-based computer-vision techniques* . Please specify the model/technique here or mention the section number about further details, it was a bit challenging to follow.

[Our response] We have reflected this comment in the Fig 2 caption: “we extract the four frames required for scoring the execution using deep learning-based models; RAFT [13], and XMem [14]. Second, from the extracted video frames, we obtain the joint coordinates of the on-screen gymnast using a pose-estimation model (BODY 25) of OpenPose [15] then calculate the joint angles in each frame. Finally, on the basis of the joint angles, we estimate the E-score deductions during Unit 2 using Random Forests [16], which enables us to automatize the feature selection by reference to the nonzero values of feature importances. More details of these three processes are in the fourth to sixth subsections of Materials and Methods (see also Figs. 4−6)” (p. 3, after line 84).

o Title of the manuscript shows authors mention selection of features. I am failing to understand that the selection was manual or automatic.

[Our response] Our proposed framework automatically works to feature selection once the videos of a wheel gymnast performing the tuck-front somersault performance are uploaded. As additionally noted in the caption of Fig. 6 (B), many features have zero feature importance values. In other words, the Random Forests automatically select a small number of motion features with nonzero feature importance values related to the E-score deductions. When further narrowing down the candidates into one or two, as in the discussion, we need to manually select features with prominent feature importance values. We have reflected this comment in the Fig 2 caption: “Finally, on the basis of the joint angles, we estimate the E-score deductions during Unit 2 using Random Forests [16], which enables us to automatize the feature selection by reference to the nonzero values of feature importances” (p. 3, after line 84) and in the Random Forests section: “For automatic feature selection to avoid significant E-score deductions, we measured each feature importance with the threshold of 0.1 of the trained model” (p. 8, lines 260 - 262). We have also changed the title to “Automatic feature selection for performing Unit 2 of vault in wheel gymnastics”.

o Abstract line 5: it should be its not his. Please proofread the whole manuscript.

[Our response] We have reflected this comment by changing the word “his” to “its” (p. 1, line 6). We have also corrected terminology throughout the manuscript.

Reviewer #2: The authors of the manuscript present a framework for analyzing the relationship between the movement features of a wheel gymnast during the mounting phase of Unit 2 of the vault event and execution (E-score) deductions, using a machine learning approach. They utilized gymnastics rules and machine learning techniques to determine the E-score deductions. According to the authors, this is the first study of its kind to quantify the E-score deductions in wheel gymnastics using a computer vision approach.

Some areas for improvement in the manuscript include increasing the number of subjects, as the results of the proposed framework would be stronger if they were based on the analysis of multiple gymnasts, allowing for statistical analysis and the application of null hypothesis to assess the significance of the method. The quality of the figures, specifically Figure 6, could also be improved by enlarging the font size of the x- and y-axes for better readability. Additionally, the manuscript's sentence structure and use of academic-style English could be improved.

[Our response] In response to your suggestion, we quickly recruited another male wheel gymnast, who had a similar physical size and gymnastic skill to the first gymnast. At the same time, we filed a research ethics application to our university and subsequently received approval for additional experiments with human subjects. We added the new approval number of 58 to the subsection of Participants in our manuscript (p. 4, line 131). We then recorded eight videos of the second gymnast performing a tuck-front somersault over three days. We merged them into a test dataset and analyzed them with the proposed framework, founding that they were also predictable with high accuracies of both R2 and RMSE, as shown in the new Fig. 6 (A).

In the new Fig. 6, we enlarged the font sizes and replaced the tree samples with easier-to-grasp ones.

Since the journal also requested that we revise our manuscript to meet PLOS ONE's style template, we holistically readjusted the sentence structure.

We have reflected this comment in the revised manuscript (p. 8, lines 273 - 275).

---

## [Decision Letter · Decision Letter 1]

10 Apr 2023

PONE-D-22-33476R1Automatic feature selection for performing Unit 2 of vault in wheel gymnasticsPLOS ONE

Dear Dr. KITAJIMA,

Thank you for submitting your manuscript to PLOS ONE. After careful consideration, we feel that it has merit but does not fully meet PLOS ONE’s publication criteria as it currently stands. Therefore, we invite you to submit a revised version of the manuscript that addresses the points raised during the review process.

Some minors revisions are still required. R2 has suggested to include statistical test to verify the claims and I concur with him.

We look forward to receiving your revised manuscript.

Kind regards,

Noman Naseer, PhD

Academic Editor

PLOS ONE

Journal Requirements:

Additional Editor Comments:

Some minors revisions are still required.

Reviewers' comments:

Reviewer's Responses to Questions

**Comments to the Author**

1. If the authors have adequately addressed your comments raised in a previous round of review and you feel that this manuscript is now acceptable for publication, you may indicate that here to bypass the “Comments to the Author” section, enter your conflict of interest statement in the “Confidential to Editor” section, and submit your "Accept" recommendation.

Reviewer #1: All comments have been addressed

Reviewer #2: (No Response)

2. Is the manuscript technically sound, and do the data support the conclusions?

Reviewer #1: Yes

Reviewer #2: Yes

3. Has the statistical analysis been performed appropriately and rigorously? 

Reviewer #1: Yes

Reviewer #2: Yes

4. Have the authors made all data underlying the findings in their manuscript fully available?

Reviewer #1: Yes

Reviewer #2: Yes

5. Is the manuscript presented in an intelligible fashion and written in standard English?

Reviewer #1: Yes

Reviewer #2: Yes

6. Review Comments to the Author

Reviewer #1: The authors have revised the paper, I am happy to accept it for the publication in current form.

I have one minor suggestion, regarding LSMTs, mention the issue in limitations. It would be beneficial for readers.

Congratulations!

Reviewer #2: I commend the authors for their hard work and commitment in improving the manuscript. They conducted further data acquisition, processing, and analysis using the proposed framework on an additional gymnast. In addition, they have enhanced the quality of the figures and the overall structure of the manuscript.

However, the statistical significance analysis still requires improvement. It is recommended that an appropriate statistical significance test or null-hypothesis test, such as Student's t-test or ANOVA, should be employed to calculate the performance of e-score to support the proposed method and claims.

7. PLOS authors have the option to publish the peer review history of their article (what does this mean?). If published, this will include your full peer review and any attached files.

Reviewer #1: **Yes: **Farzan M. Noori

Reviewer #2: **Yes: **Syed Hammad Nazeer Gilani

---

## [Author Response · Author response to Decision Letter 1]

12 May 2023

Academic Editor comments

Thank you for submitting your manuscript to PLOS ONE. After careful consideration, we feel that it has merit but does not fully meet PLOS ONE’s publication criteria as it currently stands. Therefore, we invite you to submit a revised version of the manuscript that addresses the points raised during the review process.

Some minors revisions are still required. R2 has suggested to include statistical test to verify the claims and I concur with him.

[Our response] Thank you for the positive comments about our manuscript. We will do our best to get it published.

Journal Requirements

 1. Please review your reference list to ensure that it is complete and correct. If you have cited

 papers that have been retracted, please include the rationale for doing so in the manuscript text,

 or remove these references and replace them with relevant current references. Any changes to

 the reference list should be mentioned in the rebuttal letter that accompanies your revised

 manuscript. If you need to cite a retracted article, indicate the article’s retracted status in the

References list and also include a citation and full reference for the retraction notice.

[Our response] We have checked that our reference list is complete and correct. For ease of finding also the article written in Japanese (reference number 12), we added the DOI to it (p. 13, lines 462-463). The following references (after number 38) in the “Revised Manuscript with Track Changes” were removed from the reference list in the “Manuscript” because they are no longer used in this study: DeepLabCut (p. 15, lines 545-547), DeeperCut (p. 15, lines 548-550), ResNet (p. 15, lines 551-553), DeepLabCut implementation (p. 15, lines 554-556), and dtreeviz

 (p. 15, lines 557-559). As for reference number 33 in the manuscript, we aligned the spacing (p.14, lines 524-525).

We also have added the following references in our manuscript after the first submission: OneFormer (p. 13, lines 483-485), Dataset repository (p. 13, lines 486-489), schematic representation vault (p. 13, lines 490-491), Comparing the Quality of Human Pose Estimation with BlazePose or OpenPose (p. 14, lines 507-509), Openpose 1.7.0 Demo (p. 14, lines 510- 513), Long short-term memory (p. 14, lines 524-525), A Multiple Comparison Procedure for Comparing Several Treatments with a Control (p. 14, lines 529-531), scipy.stats.dunnett (p. 14, lines 532-535), On a Test of Whether one of Two Random Variables is Stochastically Larger than the Other (p. 14, lines 536-538), scipy.stats.mannwhitneyu (p. 15, lines 539-542).

Responses to the review comments

Reviewer #1: The authors have revised the paper, I am happy to accept it for the publication in current form.

I have one minor suggestion, regarding LSMTs, mention the issue in limitations. It would be beneficial for readers.

Congratulations!

[Our response] Thank you for your suggestion. We have added the LSTM issue in limitations of our manuscript (p. 11-12, lines 413-420).

Reviewer #2: I commend the authors for their hard work and commitment in improving the manuscript. They conducted further data acquisition, processing, and analysis using the proposed framework on an additional gymnast. In addition, they have enhanced the quality of the figures and the overall structure of the manuscript.

However, the statistical significance analysis still requires improvement. It is recommended that an appropriate statistical significance test or null-hypothesis test, such as Student's t-test or ANOVA, should be employed to calculate the performance of e-score to support the proposed method and claims.

[Our response] In response to your suggestion, we conducted several statistical significance tests and reflected the results in Discussion (p. 10, lines 337-373), the new Fig. 7 (p. 10, after line 373), and S2 Fig (p. 15, lines 551-559).

We confirmed the statistical analysis supported our claims from the proposed methods based on the machine-learning algorithms. The previous Fig. 7 was accordingly changed to Fig 8 (p. 11, after line 401), and the content is the same.

---

## [Decision Letter · Decision Letter 2]

30 May 2023

Automatic feature selection for performing Unit 2 of vault in wheel gymnastics

PONE-D-22-33476R2

Dear Dr. KITAJIMA,

We’re pleased to inform you that your manuscript has been judged scientifically suitable for publication and will be formally accepted for publication once it meets all outstanding technical requirements.

Kind regards,

Noman Naseer, PhD

Academic Editor

PLOS ONE

Additional Editor Comments (optional):

Well revised

Reviewers' comments:

Reviewer's Responses to Questions

**Comments to the Author**

1. If the authors have adequately addressed your comments raised in a previous round of review and you feel that this manuscript is now acceptable for publication, you may indicate that here to bypass the “Comments to the Author” section, enter your conflict of interest statement in the “Confidential to Editor” section, and submit your "Accept" recommendation.

Reviewer #2: All comments have been addressed

2. Is the manuscript technically sound, and do the data support the conclusions?

Reviewer #2: Yes

3. Has the statistical analysis been performed appropriately and rigorously? 

Reviewer #2: Yes

4. Have the authors made all data underlying the findings in their manuscript fully available?

Reviewer #2: Yes

5. Is the manuscript presented in an intelligible fashion and written in standard English?

Reviewer #2: Yes

6. Review Comments to the Author

Reviewer #2: The Authors have addressed all of my concerns with the original manuscript. The revised manuscript is ready for publication.

7. PLOS authors have the option to publish the peer review history of their article (what does this mean?). If published, this will include your full peer review and any attached files.

Reviewer #2: **Yes: **Syed Hammad Nazeer Gilani

---

## [Editor Report · Acceptance letter]

15 Jun 2023

PONE-D-22-33476R2 

Automatic feature selection for performing Unit 2 of vault in wheel gymnastics 

Dear Dr. Kitajima:

I'm pleased to inform you that your manuscript has been deemed suitable for publication in PLOS ONE. Congratulations! Your manuscript is now with our production department. 

Kind regards, 

on behalf of

Dr. Noman Naseer 

Academic Editor

PLOS ONE